# Prevalence, patterns, and factors associated with tobacco use among patients with priority tobacco related illnesses at four Kenyan national referral hospitals, 2022

**Valerian Mwenda**[1]*, **Lazarus Odeny**[2], **Shukri Mohamed**[3], **Gladwell Gathecha**[1], **Anne Kendagor**[1], **Dorcas Kiptui**[1], **Florence Jaguga**[4], **Beatrice Mugi**[5], **Caroline Mithi**[6], **Kennedy Okinda**[7], **Daniel Mwai**[8], **David Njuguna**[9], **Winnie Awuor**[10], **Rachel Kitonyo-Devotsu**[10], **Jane Rahedi Ong'ang'o**[2]

**1** Department of Non-communicable Diseases, Ministry of Health, Nairobi, Kenya, **2** Kenya Medical Research Institute, Nairobi, Kenya, **3** African Population and Health Research Center, Nairobi, Kenya, **4** Moi Teaching and Referral Hospital, Eldoret, Kenya, **5** Kenyatta National Hospital, Nairobi, Kenya, **6** Kenyatta University Teaching, Referral and Research Hospital, Nairobi, Kenya, **7** Kenyatta National Hospital-Othaya Referral Hospital, Othaya, Kenya, **8** University of Nairobi, Nairobi, Kenya, **9** Department of Planning and Health Financing, Ministry of Health, Nairobi, Kenya, **10** Development Gateway, Nairobi, Kenya

* valmwenda@gmail.com

## Abstract

Tobacco use is a risk factor for many chronic health conditions. Quantifying burden of tobacco use among people with tobacco-related illnesses (TRI) can strengthen cessation programs. This study estimated prevalence, patterns and correlates of tobacco use among patients with TRI at four national referral hospitals in Kenya. We conducted a cross-sectional study among patients with five TRI (cancer, cardiovascular diseases, cerebrovascular disease, chronic obstructive pulmonary disease, and pulmonary tuberculosis) during January–July 2022. Cases identified from medical records were interviewed on socio-demographic, tobacco use and cessation information. Descriptive statistics were used to characterize patterns of tobacco use. Multiple logistic regression models were used to identify associations with tobacco use. We identified 2,032 individuals with TRI; 46% (939/2,032) had age ≥60 years, and 61% (1,241/2,032) were male. About 45% (923/2,032) were ever tobacco users (6% percent current and 39% former tobacco users). Approximately half of smokers and 58% of smokeless tobacco users had attempted quitting in the last month; 42% through cessation counselling. Comorbidities were present in 28% of the participants. Most (92%) of the patients had been diagnosed with TRI within the previous five years. The most frequent TRI were oral pharyngeal cancer (36% [725/2,032]), nasopharyngeal cancer (12% [246/2.032]) and lung cancer (10% [202/2,032]). Patients >60 years (aOR 2.24, 95% CI: 1.84, 2.73) and unmarried (aOR 1.21, 95% CI: 1.03, 1.42) had higher odds of tobacco use. Female patients (aOR 0.35, 95% CI: 0.30, 0.41) and those with no history of alcohol use (aOR 0.27, 95% CI: 0.23, 0.31), had less odds of tobacco use. Our study shows high prevalence of tobacco use among patients with TRI in Kenya, especially among older, male, less educated, unmarried, and alcohol users. We recommend tobacco use screening and cessation programs among patients with TRI as part of clinical care.

**Data Availability Statement:** The data underlying our findings has been provided as a supporting Information file (S1).

**Funding:** The work was supported by the Development Gateway to RD, through a grant from Bill & Melinda Gates Foundation. The funders had no role in study design, data collection and analysis, decision to publish, or preparation of the manuscript.

**Competing interests:** The authors have declared that no competing interests exist.

## Background

Tobacco use is a recognized risk factor for many chronic health conditions [1, 2]. Health conditions with an established link to tobacco use include cancers [3–6], cardiovascular diseases [7–12], chronic obstructive pulmonary disease (COPD) [13–15], tuberculosis [16–25] and pneumonia [26, 27]. Tobacco smoking is associated with a known 70% increase in risk of adverse health consequences and is also considered as a main risk factor leading to death globally [28]. Diseases related to tobacco use were ranked 2nd highest among all diseases burdening people globally in 2015 [3]. Overall, 7.69 million deaths and 200 million disability-adjusted life years (DALYS) were linked to smoking tobacco use in 2019 globally [29]. This accounted for 13.6% of all deaths and 7.9% of all DALYS for that year. Specifically, tobacco use has been linked to 24% and 44% of DALYs caused by Ischemic heart disease (IHD) and chronic obstructive pulmonary disease (COPD), respectively [30–32].

Current smoking contributed to the largest proportion of deaths out of the 87 risk factors included in the Global Burden of Disease study 2019 [33]. In the same year (2019), the Years of Life Lost (YLL) exceeded the Years Lived with Disability (YLDS) due to smoking tobacco use, with 168 million YLL compared to 31.6 million YLD. A higher Social Demographic Index (SDI) level has been associated with a decrease in the YLL to YLD Ratio. The lower the ratio, the greater the proportion of individuals living with chronic health illnesses due to tobacco smoking. Ischaemic heart disease, COPD, tracheal, bronchus and lung cancer, and stroke together accounted for approximately 72% of all deaths attributable to smoking tobacco use in both sexes out of the 36 health outcomes related to tobacco use [32].

The World Health Organisation (WHO) currently estimates that 8 million people die each year from tobacco use and a majority (80%) of them are from low and middle-income countries [34]. Unless countries take strict tobacco control measures, the annual burden of tobacco is expected to continue rising and will account for approximately 10% of all global deaths by 2030. Unfortunately, despite the documented risk of disease or death from tobacco related illnesses (TRI), the global burden of tobacco use remains very high. The number of current smokers globally is estimated at 1.3 billion, representing 22% of the world's population [35]. The Kenya STEPwise approach to NCD risk factor surveillance (STEPS) survey of 2015 reported the country prevalence of tobacco use as 13.3% with a male use of 23% and female prevalence of 4% [36]. Kenya has had both successes and obstacles to tobacco control, but it remains a priority health agenda in the country [37].

Even though tobacco consumption is a significant risk factor for both communicable and non-communicable diseases, there is limited information on tobacco use habits among individuals suffering from these illnesses in Kenya. Such information would be valuable to guide both policy and clinical teams, especially in tobacco cessation programs. Therefore, we sought to assess the prevalence, patterns and correlates of tobacco use among patients with at least one of five major tobacco-related illnesses (TRI), at four national referral hospitals in Kenya.

## Methods

### Study design and study sites

This was a cross-sectional study conducted at the four national referral hospitals of Kenya namely Kenyatta National Hospital (KNH), Moi Teaching and Referral Hospital (MTRH), Kenyatta National Hospital-Othaya Referral Hospital (KNH-ORH), and Kenyatta University Teaching, Referral and Research Hospital (KUTRRH). These referral hospitals provide out-patient and in-patient services to a wide population country-wide and attend to patients with the four study conditions (Cancer, CVD, cerebrovascular accidents (CVA), COPD, and pulmonary tuberculosis).

## Study population

These were patients with tobacco related illnesses (TRIs) receiving treatment as out-patients or in-patients at the four national referral hospitals. The TRIs included cancers (oral-pharyngeal, laryngeal, lung, nasopharyngeal, oesophageal), CVDs (myocardial infarction, peripheral arterial diseases), CVA, COPD (chronic bronchitis, emphysema) and pulmonary tuberculosis (PTB).

Patients were eligible for the study if they were 18 years and above, had at least one of the selected tobacco related illnesses, and had presented to the hospital for treatment. Non-consenting patients and those with severe incapacitation were considered not eligible to participate in the study.

## Sample size determination and sampling

The sample size was computed using the finite population sample size calculation formula and assuming a prevalence of tobacco use of 13% among patients with tobacco-related illnesses admitted to hospitals [38] and a non-response rate of 85%. We made an adjustment for age groups that were considered likely to have onset of cigarette smoking attributable illnesses (4 categories: 35–54, 55–64, 65–74, and 75+ [1], number of study facilities (4 facilities) and gender (2 categories). The total sample size calculated was 1845. The sample size per facility was proportionately distributed according to their workload based on bed capacity. The study factored in the annual workload by disease category and facility for the year 2020 to assign sample size by diseases condition in each facility (Table 1)

Apportioning of sample size by out-patient (OP) and in-patient (IP) was based on the ratio of annual OP workload against IP workload for each hospital. This ratio was applied on each of the condition sample sizes, except for tuberculosis that is usually managed as out-patient (Table 1).

## Sampling procedure

This study used purposive sampling as it targeted patients with TRI. Patient records, i.e., either outpatient and inpatient files or registers, were first checked to select patients with TRI. The inpatient and outpatient registers were reviewed daily in all the relevant specialty clinics (e.g., oncology and TB clinics), surgery and medical wards during the study period to identify patients who met the inclusion criteria.

## Variables

The history of tobacco use (smoking or smokeless tobacco) among patients with tobacco-related illnesses was the primary outcome variable. The status of tobacco use was reported as: never user, current user (daily user or less than daily user) and former user. Age of taking up

**Table 1. Distribution of sample size by hospital and disease condition.**

| | Annual (2020) workload by disease condition (% workload) | | | | Sample size allocation | | | | |
|---|---|---|---|---|---|---|---|---|---|
| Facility | Cancers | CVD | COPD | 673 (6%) | Cancers | CVD | COPD | TB | Total |
| KNH | 8,100 (78%) | 1,144 (11%) | 459 (4%) | 325 (7%) | 678 | 96 | 38 | 56 | 868 |
| MTRH | 3,790 (82%) | 203 (4%) | 299 (6%) | 0 (7%) | 406 | 22 | 32 | 35 | 494 |
| KUTRRH | 0 (82%) | 0 (4%) | 0 (6%) | 23 (2%) | 258 | 14 | 20 | 22 | 314 |
| KNH-ORH | 150 (12%) | 600 (47%) | 494 (39%) | 1,021 (6%) | 20 | 80 | 66 | 3 | 169 |
| Total | 12,040 (74%) | 1,947 (12%) | 1,252 (8%) | 673 (6%) | 1,366 | 221 | 142 | 116 | 1,845 |

tobacco use, and duration of its use was reported. Questions for tobacco use were adapted from the STEPs survey data collection tool [39]. Individual variables included the study TRIs that all eligible patients were enrolled with. In situations that a patient had multiple TRIs the first TRI to be diagnosed was the primary diagnosis and the other conditions were recorded as co-morbidities. Any other co-morbidity reported by the patients was also established. Socio-demographic variables include age, gender, residence (urban/rural), employment status, level of education, marital status. The respondents also reported on the history of past or current alcohol consumption, defined as intake of any alcoholic beverage, of whichever type and any amount.

Diagnosis of the target conditions was based on documented clinical and diagnostic confirmatory information (bacteriology for PTB; histology for cancer and imaging for CVD). However, for COPD, only clinical presentation information was used since lung function tests records were not available.

The definition of tobacco use status applied to both tobacco smoking and the use of smokeless tobacco. Current tobacco users were individuals who were currently using a tobacco product; a current daily tobacco user was an individual who was currently using a tobacco product daily. In this instance daily meant using a tobacco product at least every day over a period of a month or more. A current less than daily tobacco user was an individual who was not a daily user, could be an occasional tobacco user (irregular frequency or rare use) and former tobacco users were individuals who were ever tobacco users and currently were not using any tobacco product.

Exposure to secondhand smoke at home was defined as individuals who reported that smoking occurs inside their homes, while exposure to secondhand smoke at work were individuals who indicated that they had been exposed to tobacco smoke at work in the past 30 days.

## Study procedures

Data collection was conducted from January to July 2022. Two or three nurses working at the data collection sites at the participating hospitals served as study research assistants. They were trained on the standardized data collection tool, study protocol and procedures. After the training, piloting of the study procedures was conducted.

Study subjects were interviewed by research assistants using a pretested questionnaire on tobacco use. Data was collected at the patient's bedside in the wards, while outpatient study participants were interviewed at the different clinics where they were located. The research assistants were consistently supervised by a research coordinator whose work was to spot check at least 5% randomly selected interviews conducted by the interviewers, looking for inconsistencies, incomplete details, and contradictory answers. If necessary, the research assistants were asked to return to the patient to clarify the information.

Before initiating the data collection, the respective hospital management teams and departments were introduced to the research by the study investigators. A visit was made to the health facility study sites to assess feasibility of conducting the study. The team sought cooperation from the hospital authorities to support implementation.

## Data management

Data collection was conducted using Kobo Toolbox (Harvard Humanitarian Initiative), which is an open-source software for collecting survey data. This allowed collection of data offline and uploading online to transmit the data to the central database. The study data IT specialist designed the databases, ensuring integrity and security of stored data through password protection and export of data for analysis. Each study participant had a unique study identifier against their records. Only key study team members had access to these data files.

## Data analysis

Statistical Application Software (SAS version 9.0, SAS Institute Inc, CA, USA) was used to compute and analyze the study responses. Categorical characteristics were summarized as frequencies and percentages, with comparison by smoking status (current smokers, former smokers, and never smokers) and disease condition with covariates using chi-square tests. Continuous variables were summarized using means, median, standard deviations, and interquartile ranges (IQR) and tested for differences by smoking status and disease condition using Kruskal-Wallis tests. The study determined the proportion of tobacco users by disease conditions. Socio-demographic variables and tobacco product use were analyzed by disease conditions. We used bivariate and multivariate logistic regression models to assess factors that are associated with tobacco use status among patients with TRI. A 95% confidence interval was applied for all analyses. Any associations were deemed significant at 0.05 or less level of significance.

## Ethical considerations

The study research protocol was reviewed and approved by the Kenyatta National Hospital-University of Nairobi ethics and research committee and Moi University/ Moi Teaching and Referral Hospital Institutional Research and Ethics Committee. The approval references were KNH-ERC/A/346 and FAN: 00039540003954, respectively. Written informed consent was obtained from each eligible participant before enrolment into the study and after thorough explanation of the risks and benefits of participating in the study.

## Results

### Characteristics of the study participants

During the study period, we collected data from 2,032 individuals from the four referral hospitals. Table 2 presents the socio-demographic, behavioral and clinical factors. Majority of the patients were aged 60+ years (46%), men (61%), had primary-level education (41%), employed (54%), married (73%). Majority of the study participants were from the Kenyatta National referral hospital (53%), had never used tobacco (54%) and were consumers of alcohol (56%). Six percent of the study participants were current tobacco users, while 40% were former users. Majority of current tobacco users (88%) were male. Among the current smokers, 68% stated that they usually smoke within 30 minutes of waking up and a similar proportion of smokeless tobacco users stated that they use the product within 30 minutes of waking up. The average number of cigarettes smoked was 7.7 sticks per day. Overall, 60% of the current tobacco users have tried to quit smoking in the past 12 months. Comorbidities were present in 28% of the study participants, with more males having comorbidities compared to females (54%). All comorbid conditions were more prevalent among males, except HIV. Most of the patients were from the outpatient departments of the hospitals (66%).

Most (92%) of the patients were diagnosed with a tobacco-related illness in the previous five years. The three most common tobacco related illnesses were oral pharyngeal cancer (36%), nasopharyngeal cancer (12%) and lung cancer (10%). Peripheral arterial disease (PAD) was the least common TRI (2%).

### Tobacco use characteristics in the study population

Among the TRI patients, the prevalence of having ever used tobacco was 45% (Table 3). The prevalence of tobacco use was highest among patients with laryngeal cancer (61%), chronic

**Table 2. Socio-demographic characteristics of the study participants.**

| Variable | Category | a. Male | | b. Female | | Total |
|---|---|---|---|---|---|---|
| Age Categories | 18–39 Years | 189 | 54% | 162 | 46% | 351 |
| | 40–60 Years | 471 | 63% | 271 | 37% | 742 |
| | 60+ Years | 581 | 62% | 358 | 38% | 939 |
| Highest level of Education | a. No formal education | 83 | 39% | 129 | 61% | 212 |
| | b. Primary | 495 | 59% | 345 | 41% | 840 |
| | c. Secondary | 429 | 68% | 198 | 32% | 627 |
| | d. Post Secondary | 227 | 66% | 116 | 34% | 343 |
| | (blank) | 7 | 70% | 3 | 30% | 10 |
| Employment status | a. Unemployed | 464 | 50% | 460 | 50% | 924 |
| | b. Employed | 771 | 70% | 330 | 30% | 1101 |
| | (blank) | 6 | 86% | 1 | 14% | 7 |
| Marital Status | a. In Union | 1035 | 70% | 453 | 30% | 1488 |
| | b. Not in Union | 205 | 38% | 336 | 62% | 541 |
| | (blank) | 1 | 33% | 2 | 67% | 3 |
| Facility | KNH | 646 | 60% | 425 | 40% | 1071 |
| | KUT | 135 | 59% | 95 | 41% | 230 |
| | MTR | 326 | 62% | 204 | 38% | 530 |
| | ORH | 134 | 67% | 67 | 33% | 201 |
| Tobacco Use | a. Current Tobacco User | 105 | 88% | 14 | 12% | 119 |
| | b. Former Tobacco User | 706 | 88% | 98 | 12% | 804 |
| | c. Never Tobacco User | 428 | 39% | 678 | 61% | 1106 |
| | (blank) | 2 | 67% | 1 | 33% | 3 |
| Tobacco Smoke | a. Current Smoker | 91 | 93% | 7 | 7% | 98 |
| | b. Former Smoker | 680 | 91% | 64 | 9% | 744 |
| | c. Never Smoker | 465 | 39% | 718 | 61% | 1183 |
| | (blank) | 5 | 71% | 2 | 29% | 7 |
| Smokeless Tobacco Use | a. Current Smokeless user | 17 | 71% | 7 | 29% | 24 |
| | b. Former Smokeless user | 84 | 65% | 46 | 35% | 130 |
| | c. Never Smokeless user | 1135 | 61% | 734 | 39% | 1869 |
| | (blank) | 5 | 56% | 4 | 44% | 9 |
| Alcohol use | a. Yes | 944 | 83% | 191 | 17% | 1135 |
| | b. No | 294 | 33% | 593 | 67% | 887 |
| | (blank) | 3 | 30% | 7 | 70% | 10 |
| Comorbidities | a. Yes | 309 | 54% | 263 | 46% | 572 |
| | b. No | 929 | 64% | 526 | 36% | 1455 |
| | (blank) | 3 | 60% | 2 | 40% | 5 |
| Hypertension | a. Yes | 203 | 53% | 180 | 47% | 383 |
| | b. No | 106 | 56% | 83 | 44% | 189 |
| | (blank) | 932 | 64% | 528 | 36% | 1460 |
| Diabetes | a. Yes | 66 | 56% | 51 | 44% | 117 |
| | b. No | 243 | 53% | 212 | 47% | 455 |
| | (blank) | 932 | 64% | 528 | 36% | 1460 |
| HIV | a. Yes | 22 | 38% | 36 | 62% | 58 |
| | b. No | 287 | 56% | 227 | 44% | 514 |
| | (blank) | 932 | 64% | 528 | 36% | 1460 |
| Heart failure | a. Yes | 12 | 52% | 11 | 48% | 23 |
| | b. No | 297 | 54% | 252 | 46% | 549 |

*(Continued)*

**Table 2.** (Continued)

| Variable | Category | a. Male | | b. Female | | Total |
|---|---|---|---|---|---|---|
| | (blank) | 932 | 64% | 528 | 36% | 1460 |
| Cancer | a. Yes | 7 | 35% | 13 | 65% | 20 |
| | b. No | 302 | 55% | 250 | 45% | 552 |
| | (blank) | 932 | 64% | 528 | 36% | 1460 |
| Other Comorbidities | a. Yes | 75 | 56% | 59 | 44% | 134 |
| | b. No | 234 | 53% | 204 | 47% | 438 |
| | (blank) | 932 | 64% | 528 | 36% | 1460 |
| Patient admission status | a. Out-patient | 807 | 60% | 536 | 40% | 1343 |
| | b. In-patient | 434 | 63% | 255 | 37% | 689 |
| Time since diagnosis of TRI | a. < 5 years | 1156 | 62% | 711 | 38% | 1867 |
| | b. 5–10 years | 50 | 53% | 45 | 47% | 95 |
| | c. > 10 years | 35 | 50% | 35 | 50% | 70 |
| TRI | 01. Myocardial Infarction | 31 | 57% | 23 | 43% | 54 |
| | 02. Cerebral Vascular Accident | 74 | 45% | 91 | 55% | 165 |
| | 03. Peripheral arterial diseas | 23 | 64% | 13 | 36% | 36 |
| | 04a. Oral-pharyngeal cancer | 459 | 63% | 266 | 37% | 725 |
| | 04b. Oesophagus cancer | 34 | 65% | 18 | 35% | 52 |
| | 05. Laryngeal cancer | 144 | 81% | 34 | 19% | 178 |
| | 06. Lung cancer | 107 | 53% | 95 | 47% | 202 |
| | 07. Chronic bronchitis | 73 | 51% | 69 | 49% | 142 |
| | 08. Emphysema | 46 | 56% | 36 | 44% | 82 |
| | 09. Tuberculosis | 100 | 67% | 50 | 33% | 150 |
| | 10. Nasopharyngeal cancer | 150 | 61% | 96 | 39% | 246 |

bronchitis (60%), emphysema (54%) and peripheral arterial disease (53%). Patients with CVA had the lowest prevalence of tobacco use, at 23%.

**Tobacco cessation among patients with TRI.** Overall, 50% (59) of current smokers and 58% (14) of smokeless tobacco users had attempted to quit in the preceding 12 months. The most common quitting approaches were counselling, which was adopted by 39% of smokers

**Table 3. Tobacco use status among patients with various tobacco related illnesses, Kenya, 2022.**

| Tobacco related illnesses | N | Tobacco use status | |
|---|---|---|---|
| | | Ever-used tobacco (n, %) | Never used tobacco (n, %) |
| Myocardial Infarction | 54 | 18, (33.3%) | 36, (66.6%) |
| Cerebral Vascular Accident | 165 | 38, (23.0%) | 127, (77.0%) |
| Peripheral arterial diseases | 36 | 19, (52.8%) | 17, (47.2%) |
| Oral-pharyngeal cancer | 725 | 357, (49.2%) | 368, (50.6%) |
| Oesophagus cancer | 52 | 22, (42.3%) | 30, (57.7%) |
| Laryngeal cancer | 178 | 108, (60.7%) | 70, (39.3%) |
| Lung cancer | 202 | 88, (43.6%) | 114, (56.4%) |
| Chronic bronchitis | 142 | 85, (59.9%) | 57, (40.1%) |
| Emphysema | 82 | 44, (53.7%) | 38, (46.3%) |
| Tuberculosis | 150 | 51, (34.0%) | 99, (66.0%) |
| Nasopharyngeal cancer | 246 | 93, (37.8%) | 153, (62.2%) |
| **Total** | **2032** | **923, (45.4%)** | **1109, (54.6%)** |

**Table 4. Tobacco use cessation approaches among current users with TRI.**

| Smoking cessation approach | Proportion who attempted to quit among current users | |
|---|---|---|
| | Smokers | Smokeless tobacco users |
| | n (%) | n (%) |
| Counselling at a cessation clinic | 23 (38.9) | 8 (57.1) |
| Nicotine replacement therapy | 4 (6.8) | 0 (0.0) |
| Other prescription medicines | 1 (1.7) | 0 (0.0) |
| Traditional medicines | 1 (1.7) | 0 (0.0) |
| Quit/telephone support lines | 2 (3.4) | 0 (0.0) |
| Unspecified | 28 (47.5) | 6 (42.9) |

and 57% of smokeless tobacco users (Table 4). Additionally, 79% of current smokers and 63% of smokeless tobacco users reported receiving advice to quit use during their last hospital visit.

**Factors associated with ever tobacco use among patients with tobacco-related illnesses.** Table 5 shows the association between tobacco use and various socio-demographic and health variables in the study population. For females patients with TRI, the odds of being tobacco users was 65% lower than that for males (aOR 0.35, 95% CI; 0.30, 0.41). An increasing positive and significant association was observed between older age and being a tobacco user. The odds of tobacco use were twice as high for TRI patients above 60 years compared to study those aged less than 40 years (aOR 2.24, 95% CI; 1.84, 2.73). Overall, a decreasing significant association of being a tobacco user was observed with increasing level of education. Patients with post-secondary education had 51% less odds of being tobacco users compared with those

**Table 5. Factors associated with tobacco use among patients with tobacco related illnesses, Kenya, 2022.**

| Characteristic | Category | N, (%) | Crude OR | p-value | 95% CI (OR) | Adjusted OR | p-value | 95% CI (OR) |
|---|---|---|---|---|---|---|---|---|
| Sex | Male | 811, (66%) | 1 | | | 1 | | |
| | Female | 112, (14%) | 0.087 | 0.0000 | (0.07, 0.11) | 0.351 | 0.000 | (0.30, 0.41) |
| Age Categories | 18–39 Years | 67, (19%) | 1 | | | 1 | | |
| | 40–60 Years | 330, (45%) | 3.372 | 0.0000 | (2.49, 4.57) | 1.307 | 0.005 | (1.08, 1.58) |
| | 60+ Years | 526, (56%) | 5.361 | 0.0000 | (3.99, 7.21) | 2.240 | 0.000 | (1.84, 2.73) |
| Highest level of Education | No formal education | 108, (51%) | 1 | | | 1 | | |
| | Primary | 409, (49%) | 0.927 | 0.6208 | (0.69, 1.25) | 1.189 | 0.090 | (0.97, 1.45) |
| | Secondary | 277, (44%) | 0.769 | 0.0986 | (0.56, 1.05) | 0.783 | 0.026 | (0.63, 0.97) |
| | Post-Secondary | 125, (37%) | 0.555 | 0.0009 | (0.39, 0.79) | 0.488 | 0.000 | (0.37, 0.64) |
| Employment status | Unemployed | 379, (41%) | 1 | | | 1 | | |
| | Employed | 538, (49%) | 1.366 | 0.0005 | (1.14, 1.63) | 0.972 | 0.684 | (0.85, 1.11) |
| Marital Status | In Union | 749, (51%) | 1 | | | 1 | | |
| | Not in Union | 172, (32%) | 0.459 | 0.0000 | (0.37, 0.57) | 1.207 | 0.021 | (1.03, 1.42) |
| Alcohol use | Yes | 820, (73%) | 1 | | | 1 | | |
| | No | 102, (12%) | 0.050 | 0.0000 | (0.04, 0.06) | 0.270 | 0.000 | (0.23, 0.31) |
| Comorbidities | Yes | 247, (43%) | 1 | | | 1 | | |
| | No | 672, (46%) | 0.933 | 0.0000 | (1.38, 0.21) | 1.113 | 0.134 | (0.97, 1.28) |
| Patient admission status | Out-patient | 589, (44%) | 1 | | | 1 | | |
| | In-patient | 334, (49%) | 1.214 | 0.0401 | (1.01, 1.46) | 1.051 | 0.455 | (0.92, 1.20) |
| Time since diagnosis of TRI | < 5 years | 865, (47%) | 1 | | | 1 | | |
| | 5–10 years | 35, (37%) | 0.669 | 0.0648 | (0.44, 1.03) | 0.656 | 0.161 | (0.36, 1.18) |
| | > 10 years | 23, (33%) | 0.561 | 0.0256 | (0.34, 0.93) | 0.426 | 0.015 | (0.21, 0.85) |

with no formal education (aOR 0.49, 95% CI; 0.37, 0.64). Patients not in marital union had 21% higher odds of being tobacco users compared with those in some form of union (aOR 1.21; 95% CI; 1.03, 1.42). Patients with no history of alcohol use had 73% less odds of being tobacco users (aOR 0.27, 95% CI; 0.23, 0.31). Patients with time since diagnosis with TRI of more than 10 years had 53% lower odds of tobacco use compared with those with less than 5 years (aOR 0.43, 95% CI; 0.21, 0.85).

## Discussion

In this study, nearly half of patients with TRI had history of tobacco use. Over half of current users had attempted quitting in the previous month and two-thirds had received quitting advice in their last hospital visit. Older age, alcohol use, male sex, not being in a marital union were associated with higher odds of tobacco use among TRI patients, while more educated patients, those with higher level of education and longer period since diagnosis of TRI had lower odds of tobacco use.

Cancers and CVA formed the largest proportion of TRIs in this hospital-based study. This is comparable to a study on health cost of tobacco use in Uganda by Nargis et al, in which 30% of patients had CVD, 27% had oral pharyngeal cancer and 14% had COPD [40]. The settings of the studies were also similar, being national referral hospitals, but the Ugandan study was conducted in specialized units in the same center. One difference is that in our study, we reported chronic bronchitis and emphysema separately, but combined as COPD constituted the third frequently reported TRI after oral and nasopharyngeal cancer.

Nearly half of the patients in our study had history of either current or past use of tobacco. This is higher than the national average tobacco use in Kenya, pointing towards higher burden of use among patients with TRI compared with the general population [36]. Although the epidemiological link between tobacco use and the various TRIs has strongly been demonstrated in the past, quantifying the burden in specific local contexts can provide additional impetus to tobacco control [41]. We observed different patterns of tobacco use among patients with certain TRI in this study. For instance, only 44% of patients with lung cancer in our study had history of ever using tobacco. This compares with another study done in Kenya, that showed only 36% of patients with lung cancer had history of current cigarette smoking [42]. This differs from the global average, where approximately 85% of lung cancer is linked to smoking [43]. Therefore, additional factors like long-term air pollution, other environmental exposures, genetic predisposition, infectious comorbidities, and other individual lifestyle factors could contribute to occurrence of TRI like lung cancer in Kenya, and warrant attention by public health authorities [44–46].

Half of current tobacco users had attempted quitting in the previous month and majority of the them had received quitting advice from healthcare workers during their last hospital visit. Most of those who attempted to quit had used counselling as the support mechanism. While this provides an opportunity for strengthening cessations programs in Kenya, a combination of various approaches adopting the 5 A's model may be more effective [47]. Integrating cessation programs in routine clinical care may be more efficient in increasing cessation among patients with TRI.

Older patients, males, those with history of alcohol use and not being in a marital union had higher odds of tobacco use in our study. Since tobacco-related morbidity and mortality manifests several decades after use initiation, burden of disease would be higher in older patients, while younger patients with TRI may have other risk factors [48]. Tobacco use prevalence is higher among males compared with females globally; therefore, burden of disease and attribution would also likely be higher among males [35]. However, this pattern may change,

as more females take-up tobacco use across the world [49]. Alcohol use may be related to the same drivers of addiction as tobacco, including social, psychological, and environmental factors [50]. Marital union may represent presence of social support mechanisms, absence of which may predispose one to tobacco and other substance use [51]. Such mechanism can be exploited to increase success of cessation support programs. A study in Ghana on tobacco use among older adults by Yawson et al showed association between tobacco use and male sex and general lack of satisfaction in life, but none with age or alcohol use; however, this study was conducted among older persons in the general population [52].

Our study has particular strengths. First, it is the first, to our knowledge, to attempt to quantify tobacco use among hospital patients with TRI in the Kenya. Therefore, the findings will be applicable in making a public health case for sustained tobacco control, especially integrating cessation counselling and support in routine clinical care. Second, the study was conducted at all national tertiary hospitals. Since, by nature most TRIs are managed at tertiary level, we are confident we captured majority of the TRIs diagnosed in the country during the period under focus. However, the study also has limitations. First, this study was hospital-based, which may not be representative of the wider population. As such the findings may not be generalized to the country population at large. Second, the case definition for the TRI was based on the already documented diagnosis and not diagnostic investigations conducted by the study team. This could have reduced the specificity of some of the cases, especially COPD in instances where spirometry readings were unavailable. To reduce the impact of the latter on the study, every effort was made to review all supporting documentation for the diagnosis documented on the patient record. Third, tobacco use assessment was done through self-report, which could have under-estimated the prevalence due to social desirability bias.

## Conclusion and recommendations

Our study highlights the burden of tobacco use among people diagnosed with priority TRI, as well as cessation and associations that can guide interventions. Estimating tobacco use among people diagnosed with TRI can enrich the local anti-tobacco advocacy toolkit. Further studies can quantify the burden of TRIs more effectively in the population and assess the efficacy of various cessation approaches in this population, to further inform anti-tobacco policy and clinical practice advocacy in Kenya.

## Supporting information

**S1 Data. De-identified study dataset.**
(XLSX)

## Acknowledgments

We acknowledge the data collectors, hospital administrators, Department of non-communicable diseases, Ministry of Health, African Population and Health Research Center and Development Gateway for their facilitation in the conduction of this study.

## Author Contributions

**Conceptualization:** Lazarus Odeny, Shukri Mohamed, Gladwell Gathecha, Anne Kendagor, Dorcas Kiptui, Daniel Mwai, Winnie Awuor, Rachel Kitonyo-Devotsu, Jane Rahedi Ong'ang'o.

**Data curation:** Valerian Mwenda, Lazarus Odeny.

**Formal analysis:** Lazarus Odeny.

**Funding acquisition:** Rachel Kitonyo-Devotsu.

**Investigation:** Valerian Mwenda, Lazarus Odeny, Shukri Mohamed, Gladwell Gathecha, Anne Kendagor, Florence Jaguga, Caroline Mithi, Kennedy Okinda, Daniel Mwai, David Njuguna, Rachel Kitonyo-Devotsu, Jane Rahedi Ong'ang'o.

**Methodology:** Valerian Mwenda, Lazarus Odeny, Shukri Mohamed, Gladwell Gathecha, Anne Kendagor, Beatrice Mugi, Kennedy Okinda, Daniel Mwai, David Njuguna, Rachel Kitonyo-Devotsu, Jane Rahedi Ong'ang'o.

**Project administration:** Shukri Mohamed, Gladwell Gathecha, Anne Kendagor, Dorcas Kiptui, Daniel Mwai, Winnie Awuor, Rachel Kitonyo-Devotsu, Jane Rahedi Ong'ang'o.

**Supervision:** Gladwell Gathecha, Anne Kendagor, Dorcas Kiptui, Florence Jaguga, Beatrice Mugi, Caroline Mithi, Kennedy Okinda, Winnie Awuor, Jane Rahedi Ong'ang'o.

**Validation:** Gladwell Gathecha.

**Writing – original draft:** Valerian Mwenda.

**Writing – review & editing:** Valerian Mwenda, Lazarus Odeny, Shukri Mohamed, Gladwell Gathecha, Anne Kendagor, Dorcas Kiptui, Florence Jaguga, Beatrice Mugi, Caroline Mithi, Kennedy Okinda, Daniel Mwai, David Njuguna, Jane Rahedi Ong'ang'o.

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

13 / 13