## [Decision Letter · Decision Letter 0]

9 Aug 2023

PGPH-D-23-00782

Prevalence and factors associated with tobacco use among patients with tobacco related illness in four national referral hospitals of Kenya

Dear Dr. Mwenda,

Thank you for submitting your manuscript to PLOS Global Public Health. After careful consideration, we feel that it has merit but does not fully meet PLOS Global Public Health’s publication criteria as it currently stands. Therefore, we invite you to submit a revised version of the manuscript that addresses the points raised during the review process.

EDITOR: Please insert comments here and delete this placeholder text when finished. Be sure to:

Please ensure that your decision is justified on PLOS Global Public Health’s publication criteria and not, for example, on novelty or perceived impact.

We look forward to receiving your revised manuscript.

Kind regards,

Dickson Abanimi Amugsi, PhD

Academic Editor

Journal Requirements:

1.Please amend your detailed Financial Disclosure statement. This is published with the article. It must therefore be completed in full sentences and contain the exact wording you wish to be published.

Additional Editor Comments (if provided):

Thank you for submitting your manuscript to PGPH for publication. Five reviewers have assessed it. Three of the reviewers recommended that it should be rejected. Reviewers 2 & 4 felt that the work has no added value, as it basically confirmed the extant literature. Further, although reviewer 1 suggested minor revision, they raised substantial issues that will need your attention.

The mixed reviews above notwithstanding, I felt the manuscript has merit, hence my decision to give you the opportunity to revise it and resubmit for consideration. Please ensure that you carefully address each and every comment from all the five reviewers, including reviewer 3 who recommended that it should be accepted for publication.

Good luck in the revision.

Reviewers' comments:

Reviewer's Responses to Questions

**Comments to the Author**

1. Does this manuscript meet PLOS Global Public Health’s publication criteria? Is the manuscript technically sound, and do the data support the conclusions? The manuscript must describe methodologically and ethically rigorous research with conclusions that are appropriately drawn based on the data presented.

Reviewer #1: Yes

Reviewer #2: Partly

Reviewer #3: Yes

Reviewer #4: Yes

Reviewer #5: Yes

2. Has the statistical analysis been performed appropriately and rigorously?

Reviewer #1: Yes

Reviewer #2: Yes

Reviewer #3: Yes

Reviewer #4: Yes

Reviewer #5: Yes

3. Have the authors made all data underlying the findings in their manuscript fully available (please refer to the Data Availability Statement at the start of the manuscript PDF file)?

Reviewer #1: Yes

Reviewer #2: Yes

Reviewer #3: Yes

Reviewer #4: Yes

Reviewer #5: Yes

4. Is the manuscript presented in an intelligible fashion and written in standard English?

Reviewer #1: Yes

Reviewer #2: Yes

Reviewer #3: Yes

Reviewer #4: Yes

Reviewer #5: Yes

5. Review Comments to the Author

Reviewer #1: Overall, the manuscript is well written and it focuses on an important public health issue, building on the existing evidence demonstrating the link between tobacco use and negative health effects. Being able to define and better understand the prevalence of use in this specific patient cohorts can help build a case for interventions for tobacco use prevention and cessation programs.

The authors can consider addressing the following comments to improve the clarity of the manuscript.

1. In the background section, the authors state "Furthermore, the findings on cost of health care for tobacco-related illnesses will help to inform stronger policy interventions that are required to minimize tobacco use and the resulting disease burden in Kenya." This is an over reach as the manuscript findings do not address any aspect of cost of health care, instead the findings focus is on prevalence of tobacco use. The authors can consider including this costing aspect or editing this specific section of the background.

2. In the background section, the authors end the section with the statement "Therefore, we sought to assess the prevalence of tobacco use among patients with at least one of four major TRI, at the four national referral hospitals in Kenya." The authors, should spell out TRI on first use in the statement. Also, this is the first time the authors introduce four national referral hospital, leaving the reader lost. Consider describing the hospitals in this section or simply dropping "the".

3. In the methods section, the authors define the outcome variable of tobacco use well. It is not clear though whether information on duration of use and amount used (pack years) was collected? Even though for tobacco use there is no threshold for toxicity when it comes to health effects, I believe a better exposure evaluation is informative in building a case on the link between tobacco and TRIs and also in developing cessation programs in this specific cohort of patients. If collected this information can be included in the results section.

4. Table 3, tabulates the comparison between ever used tobacco and never used tobacco as it relates to the various TRIs, and table 4 shows factors associated with ever used tobacco in patients with tobacco related illnesses. While this is statistically sound, from a clinical perspective it is troublesome lumping an individual who has perhaps smoked one puff of a cigarette in their lifetime with one who has smoked 1 pack per day for a year. Was there any attempt made to measure the amount a person has smoked over a specified period of time?

5. The authors state that reporting ever used tobacco was associated with increasing age. Wouldn't this naturally be expected with a measure of ever used.

6. Perhaps for preventive programs a broad measure of ever used may be helpful, however for cessation program intervention, I believe current use measure is more critical in establishing burden and resource planning for cessation interventions.

7. In the conclusion, the authors conclude with a statement "Our study highlights the real-world health implications of tobacco use in Kenya". This statement is an over reach as this was a cross sectional survey showing associations and not causality. The statement suggests causality of TRIs was established in the study.

8. The authors can consider specifically including the need for cessation support programs for these cohort of patients in addition to anti-tobacco advocacy efforts mentioned in the conclusion.

9. Overall, the lack of granularity in the classification of tobacco use history, misses the opportunity to better understand the burden and needs of these cohort of patients. The study authors merely classify use as current, former and never without bringing out aspects of duration, frequency and quantity of tobacco use.

10. Lastly, I believe there has already been great strides towards anti tobacco advocacy and smoking cessation in Kenya and this has already translated in a lowered overall prevalence of tobacco use in the country. The authors can consider including this information in the background.

Reviewer #2: Thank you for the opportunity to review this paper on the prevalence and factors associated with tobacco use among patients with tobacco-related illness in four national referral hospitals of Kenya. This is an important subject, particularly because it provides information for tobacco control policies in a lower-middle-income country with a high burden of tobacco use and limited treatment resources. My major concern is regarding the study's results, which predominantly confirm well-known findings in the existing literature. For instance, factors such as being female, married, having higher education, and having no history of alcohol use were already established as predictors of abstaining from tobacco use. However, the methods and the paper overall are very well written and may be more suitable for a local journal.

Reviewer #3: the data description, analysis and write up are up to standard according to my view points .i am concerned tat using a purposive sampling method for this type of data might make the results prone to bias,

Reviewer #4: The authors conducted a cross-sectional study linking tobacco use among patients with tobacco-related diseases. As such, there is no new information from this study. Similar published studies are already available (https://pubmed.ncbi.nlm.nih.gov/30455795/, https://pubmed.ncbi.nlm.nih.gov/30400915/). Rather, as the risk factors are already known, interventions around reducing tobacco use and demonstrating its effect would be more beneficial. Authors can try for a regional journal for their manuscript.

Reviewer #5: Thank you for the study.

The title of the is confusing and not provide a clear picture of the study conducted.

Objectives of the study and justification for the study is not clear.

Even though authors have collected a significant amount of data, they are not logically presented.

Discussion is very superficial and is mostly descriptive.

There is no meaningful comparison with the available literatures, explanation's for findings and discussion.

Limitations of the study needs to be highlighted.

6. PLOS authors have the option to publish the peer review history of their article (what does this mean?). If published, this will include your full peer review and any attached files.

**Do you want your identity to be public for this peer review?** For information about this choice, including consent withdrawal, please see our Privacy Policy.

Reviewer #1: No

Reviewer #2: No

Reviewer #3: No

Reviewer #4: No

Reviewer #5: No

---

## [Decision Letter · Decision Letter 1]

16 Oct 2023

Prevalence, patterns, and factors associated with tobacco use among patients with priority tobacco related illnesses at four Kenyan national referral hospitals, 2022

PGPH-D-23-00782R1

Dear Mwenda,

We are pleased to inform you that your manuscript 'Prevalence, patterns, and factors associated with tobacco use among patients with priority tobacco related illnesses at four Kenyan national referral hospitals, 2022' has been provisionally accepted for publication in PLOS Global Public Health.

Best regards,

Dickson Abanimi Amugsi, PhD

Academic Editor

Reviewer Comments (if any, and for reference):

Reviewer's Responses to Questions

**Comments to the Author**

1. If the authors have adequately addressed your comments raised in a previous round of review and you feel that this manuscript is now acceptable for publication, you may indicate that here to bypass the “Comments to the Author” section, enter your conflict of interest statement in the “Confidential to Editor” section, and submit your "Accept" recommendation.

Reviewer #1: All comments have been addressed

2. Does this manuscript meet PLOS Global Public Health’s publication criteria? Is the manuscript technically sound, and do the data support the conclusions? The manuscript must describe methodologically and ethically rigorous research with conclusions that are appropriately drawn based on the data presented.

Reviewer #1: Yes

3. Has the statistical analysis been performed appropriately and rigorously?

Reviewer #1: Yes

4. Have the authors made all data underlying the findings in their manuscript fully available (please refer to the Data Availability Statement at the start of the manuscript PDF file)?

Reviewer #1: Yes

5. Is the manuscript presented in an intelligible fashion and written in standard English?

Reviewer #1: Yes

6. Review Comments to the Author

Reviewer #1: My comments have been adequately addressed.

7. PLOS authors have the option to publish the peer review history of their article (what does this mean?). If published, this will include your full peer review and any attached files.

**Do you want your identity to be public for this peer review?** For information about this choice, including consent withdrawal, please see our Privacy Policy.

Reviewer #1: No
